# Connecting Sphere Manifolds Hierarchically for Regularization

## Abstract

This paper considers classification problems with hierarchically organized classes. We force the classifier (hyperplane) of each class to belong to a sphere manifold, whose center is the classifier of its super-class. Then, individual sphere manifolds are connected based on their hierarchical relations. Our technique replaces the last layer of a neural network by combining a spherical fully-connected layer with a hierarchical layer. This regularization is shown to improve the performance of widely used deep neural network architectures (ResNet and DenseNet) on publicly available datasets (CIFAR100, CUB200, Stanford dogs, Stanford cars, and Tiny-ImageNet).

## 1    Introduction

Applying *inductive biases* or prior knowledge to inference models is a popular strategy to improve their generalization performance (Battaglia et al., 2018). For example, a hierarchical structure is found based on the similarity or shared characteristics between samples and thus becomes a basic criterion to categorize particular objects. The *known* hierarchical structures provided by the datasets (e.g., ImageNet (Deng et al., 2009) classified based on the WordNet graph; CIFAR100 (Krizhevsky, 2009) in ten different groups) can help the network identify the similarity between the given samples.

In classification tasks, the final layer of neural networks maps embedding vectors to a discrete target space. However, there is no mechanism forcing similar categories to be distributed close to each other in the embedding. Instead, we may observe classes to be uniformly distributed after training, as this simplifies the separation by the last fully-connected layer. This behavior is a consequence of seeing the label structure as 'flat,' i.e., when we omit to consider the hierarchical relationships between classes (Bilal et al., 2017).

To alleviate this problem, in this study, we force similar classes to be closer in the embedding by forcing their hyperplanes to follow a given hierarchy. One way to realize that is by making children nodes dependent on parent nodes and constraining their distance through a regularization term. However, the norm itself does not give a relevant information on the closeness between classifiers. Indeed, two classifiers are close if they classify two similar points in the same class. This means similar classifiers have to indicate a similar direction. Therefore, we have to focus on the *angle* between classifiers, which can be achieved through spherical constraints.

**Contributions.** In this paper, we propose a simple strategy to incorporate hierarchical information in deep neural network architectures with minimal changes to the training procedure, by *modifying only the last layer*. Given a hierarchical structure in the labels under the form of a tree, we explicitly force the classifiers of classes to belong to a sphere, whose center is the classifier of their super-class, recursively until we reach the root (see Figure 2). We introduce the *spherical fully-connected layer* and the *hierarchically connected layer*, whose combination implements our technique. Finally, we investigate the impact of Riemannian optimization instead of simple norm normalization.

By its nature, the proposed technique is quite versatile because the modifications only affect the structure of last fully-connected layer of the neural network. Thus, it can be combined with many other strategies (like spherical CNN from Xie et al. (2017), or other deep neural network architectures).

**Related works.** Hierarchical structures are well-studied, and their properties can be effectively learned using manifold embedding. The design of the optimal embedding to learn the latent hierarchy

is a complex task, and was extensively studied in the past decade. For example, Word2Vec (Mikolov et al., 2013b;a) and Poincaré embedding (Nickel & Kiela, 2017) showed a remarkable performance in hierarchical representation learning. (Du et al., 2018) forced the representation of sub-classes to "orbit" around the representation of their super-class to find similarity based embedding. Recently, using elliptical manifold embedding (Batmanghelich et al., 2016), hyperbolic manifolds (Nickel & Kiela, 2017; De Sa et al., 2018; Tifrea et al., 2018), and a combination of the two (Gu et al., 2019; Bachmann et al., 2019), shown that the latent structure of many data was non-Euclidean (Zhu et al., 2016; Bronstein et al., 2017; Skopek et al., 2019). (Xie et al., 2017) showed that spheres (with angular constraints) in the hidden layers also induce diversity, thus reducing over-fitting in latent space models.

Mixing hierarchical information and structured prediction is not new, especially in text analysis (Koller & Sahami, 1997; McCallum et al., 1998; Weigend et al., 1999; Wang et al., 1999; Dumais & Chen, 2000). Partial order structure of the visual-semantic hierarchy is exploited using a simple order pair with max-margin loss function in (Vendrov et al., 2016). The results of previous studies indicate that exploiting hierarchical information during training gives better and more resilient classifiers, in particular when the number of classes is large (Cai & Hofmann, 2004). For a given hierarchy, it is possible to design structured models incorporating this information to improve the efficiency of the classifier. For instance, for support vector machines (SVMs), the techniques reported in (Cai & Hofmann, 2004; 2007; Gopal et al., 2012; Sela et al., 2011) use hierarchical regularization, forcing the classifier of a super-class to be close to the classifiers of its sub-classes. However, the intuition is very different in this case, because SVMs do not learn the embedding.

In this study, we consider that the hierarchy of the class labels is known. Moreover, we *do not* change prior layers of the deep neural network, and only work on the last layer that directly contributed to build hyperplanes for a classification purpose. Our work is thus orthogonal to those works on embedding learning, but not incompatible.

**Comparison with hyperbolic/Poincaré/graph networks.** Hyperbolic network is a recent technique that shows impressive results for hierarchical representation learning. Poincaré networks (Nickel & Kiela, 2017) were originally designed to *learn* the latent hierarchy of data using *low-dimension* embedding. To alleviate their drawbacks due to a transductive property which cannot be used for unseen graph inference, hyperbolic neural networks equipped set aggregation operations have been proposed (Chami et al., 2019; Liu et al., 2019). These methods have been mostly focused on learning embedding using a hyperbolic activation function for hierarchical representation. Our technique is orthogonal to these works: First, we assume that the hierarchical structure is not learnt but already known. Second, our model focuses on generating individual hyperplanes of embedding vectors given by the network architecture. While spherical geometry has a positive curvature, moreover, that of hyperbolic space has a constant negative curvature. However, our technique and hyperbolic networks *are not mutually exclusive*. Meanwhile focusing on spheres embedded in $\mathbb{R}^d$ in this study, it is straightforward to consider spheres embedded in hyperbolic spaces.

## 2 HIERARCHICAL REGULARIZATION

### 2.1 DEFINITION AND NOTATIONS

We assume we have samples with hierarchically ordered classes. For instance, apple, banana, and orange are classes that may belong to the super-class "fruits." This represents hierarchical relationships with trees, as depicted in Figure 1.

We identify nodes in the graph through the path taken in the tree. To represent the leaf (highlighted in blue in Figure 1), we use the notation $n_{\{1,3,2\}}$. This means it is the second child of the super-class $n_{\{1,3\}}$, and recursively, until we reach the root.

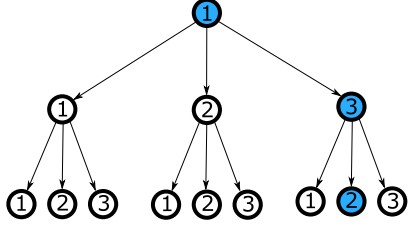

Figure 1: To reference the node at the bottom, we use the notation $n_p$ with $p = \{1, 3, 2\}$. We use curly brackets {} to write a path, and angle brackets $\langle \cdot \rangle$ for the concatenation of paths.

More formally, we identify nodes as $n_p$, where $p$ is the path to the node. A path uniquely defines a node where only one possible path exists. Using the concatenation, between the path $p$ and its child $i$, a new path $\tilde{p}$ can be defined as follows,

$$\tilde{p} = \langle p, i \rangle \tag{1}$$

We denote $\mathcal{P}$ the set of all paths in the tree starting from the root, with cardinality $|\mathcal{P}|$. Notice that $|\mathcal{P}|$ is also the number of nodes in the tree (i.e., number of classes and super-classes). We distinguish the set $\mathcal{P}$ from the set $\mathcal{L}$, the set of paths associated to nodes whose label appears in the dataset. Although $\mathcal{L}$ may equal to $\mathcal{P}$, this is not the case in our experiments. We show an example in Appendix A.

## 2.2 SIMILARITY BETWEEN OBJECTS AND THEIR REPRESENTATION

Let $X$ be the network input (e.g. an image), and $\phi_\theta(X)$ be its representation, i.e., the features of $X$ extracted by a deep neural network parameterized by $\theta$. We start with the following observation:

*Given a representation, super-class separators should be similar to separators for their sub-classes.*

This assumption implies the following direct consequence.

*All objects whose labels belong to the same super-class have a similar representation.*

That is a natural property that we may expect from a good representation. For instance, two dogs from different breeds should share more common features than that of a dog shares with an apple. Therefore, the parameter of the classifiers that identify dog's breed should also be similar. Their difference lies in the parameters associated to some specific features that differentiate breeds of dogs.

Although this is not necessarily satisfied with *arbitrary* hierarchical classification, we observe this in many existing datasets. For instance, Caltech-UCSD Birds 200 and Stanford dogs are datasets that classify, respectively, birds and dogs in term of their breeds. A possible example where this assumption may not be satisfied is a dataset whose super-classes are "labels whose first letter is «·»."

## 2.3 HIERARCHICAL REGULARIZATION

Starting from a simple observation in the previous section, we propose a regularization technique that forces the network to have similar representation for classes along a path $p$, which implies having similar representation between similar objects. More formally, if we have an optimal classifier $w_p$ for the super-class $p$ and a classifier $w_{\langle p,i \rangle}$ for the class $\langle p, i \rangle$, we expect that

$$\|w_p - w_{\langle p,i \rangle}\| \quad \text{is small.} \tag{2}$$

If this is satisfied, separators for objects in the same super-class are also similar because

$$\|w_{\langle p,i \rangle} - w_{\langle p,j \rangle}\| = \|(w_{\langle p,i \rangle} - w_p) - (w_{\langle p,j \rangle} - w_p)\| \leq \underbrace{\|w_p - w_{\langle p,i \rangle}\|}_{\text{small}} + \underbrace{\|w_p - w_{\langle p,j \rangle}\|}_{\text{small}}. \tag{3}$$

However, the optimal classifier for an arbitrary representation $\phi_\theta(X)$ may not satisfy equation 2. The naive and direct way to ensure equation 2 is through *hierarchical regularization*, which forces classifiers in the same path to be close to each other.

## 2.4 HIERARCHICAL LAYER AND HIERARCHICALLY CONNECTED LAYER

In the previous section, we described the hierarchical regularization technique given a hierarchical structure in the classes. In this section, we show how to conveniently parametrize equation 2. We first express the classifier as a sum of vectors $\delta$ defined recursively as follows:

$$w_{\langle p,i \rangle} = w_p + \delta_{\langle p,i \rangle}, \quad \delta_{\{\}} = \mathbf{0}, \tag{4}$$

where $\{\}$ is the root. It is possible to consider $\delta_{\{\}} \neq \mathbf{0}$, which shifts separating hyper-planes. We do not consider this case in this paper. Given equation 4, we have that $\|\delta_{\langle p,i \rangle}\|$ is small in equation 2. Finally, it suffices to penalize the norm of $\delta_{\langle p,i \rangle}$ during the optimization. Notice that, by construction, the number of $\delta$'s is equal to the number of nodes in the hierarchical tree.

Next, consider the output of CNNs for classification,

$$\phi_\theta(\cdot)^T W, \tag{5}$$

where $\theta$ denotes the parameters of the hidden layers, $W = [w_1, \ldots, w_{|\mathcal{L}|}]$ denotes the last fully-connected layer, and $w_i$ denotes the separator for the class $i$. For simplicity, we omit potential additional nonlinear functions, such as a softmax, on top of the prediction.

We have parametrized $w_i$ following the recursive formula in equation 4. To define the matrix formulation of equation 4, we first introduce the *Hierarchical layer* $\mathbf{H}$ which plays an important role. This hierarchical layer can be identified to the adjacency matrix of the hierarchical graph.

**Definition 1. (Hierarchical layer).** Consider ordering over the sets $\mathcal{P}$ and $\mathcal{L}$, i.e., for $i = 1, \ldots, |\mathcal{P}|$ and $j = 1, \ldots, |\mathcal{L}|$,

$$\mathcal{P} = \{p_1, \ldots, p_i, \ldots, p_{|\mathcal{P}|}\} \quad \text{and} \quad \mathcal{L} = \{p_1, \ldots, p_j, \ldots, p_{|\mathcal{L}|}\}.$$

In other words, we associate to all nodes an index. Then, the hierarchical layer $\mathbf{H}$ is defined as

$$\mathbf{H} \in \mathbb{B}^{|\mathcal{P}| \times |\mathcal{L}|}, \quad \mathbf{H}_{i,j} = 1 \text{ if } n_{p_i} \preceq n_{p_j}, \ 0 \text{ otherwise.} \tag{6}$$

where $n_{p_i} \preceq n_{p_j}$ means $n_{p_j}$ is a parent of $n_{p_i}$.

We illustrate an example of $\mathbf{H}$ in Appendix A. The next proposition shows that equation 5 can be written using a simple matrix-matrix multiplication, involving the hierarchical layer.

**Proposition 1.** Consider a representation $\phi_\theta(\cdot)$, where $\phi_\theta(\cdot) \in \mathbb{R}^d$. Let $W$ be the matrix of separators

$$W = [w_{p_1}, \ldots, w_{p_{|\mathcal{L}|}}], \quad p_i \in \mathcal{L}, \tag{7}$$

where the separators are parametrized as equation 4. Let $\Delta$ be defined as

$$\Delta \in \mathbb{R}^{d \times |\mathcal{P}|}, \quad \Delta = [\delta_{p_1}, \ldots, \delta_{p_{|\mathcal{P}|}}], \tag{8}$$

where $\mathcal{P}$ and $\mathcal{L}$ are defined in Section 2.1. Consider the hierarchical layer defined in Definition 1. Then, the matrix of separators $W$ can be expressed as

$$W = \Delta \mathbf{H}. \tag{9}$$

We can see $W = \Delta \mathbf{H}$ as a combination of an augmented fully-connected layer, combined with the hierarchical layer that selects the right columns of $\Delta$, hence the term *hierarchically connected layer*. The $\ell_2$ regularization of the $\delta$ can be conducted by the parameter *weight decay*, which is widely used in training of neural networks. The hierarchical layer $\mathbf{H}$ is fixed, while $\Delta$ is learnable. This does not affect the complexity of the back-propagation significantly, as $\Delta \mathbf{H}$ is a simple linear form.

The size of the last layer slightly increases, from $|\mathcal{L}| \times d$ to $|\mathcal{P}| \times d$, where $d$ is the dimension of the representation $\phi_\theta(\cdot)$. For instance, in the case of Tiny-ImageNet, the number of parameters *of the last layer only* increases by roughly $36\%$; nevertheless, the increased number of parameters of the last layer is still usually negligible in comparison with the total number of parameters for classical network architectures.

## 3  HIERARCHICAL SPHERES

The *hierarchical ($\ell_2$) regularization* introduced in the previous section induces separated hyper-planes along a path to be close to each other. However, this approach has a significant drawback.

We rewind equation 2, which models the similarity of two separators $w_p$ and $w_{\langle p, i \rangle}$. The similarity between separators (individual hyper-planes) should indicate that they point roughly the same direction, i.e.,

$$\left\| \frac{w_p}{\|w_p\|} - \frac{w_{\langle p, i \rangle}}{\|w_{\langle p, i \rangle}\|} \right\| \quad \text{is small.} \tag{10}$$

However, this property is *not* necessarily captured by equation 2. For instance, assume that $w_p = -w_{\langle p, i \rangle}$, i.e., the separators point in two opposite directions (and thus completely different). Then, equation 2 can be arbitrarily small in the function of $\|w_p\|$ but not in equation 10:

$$\|w_p - w_{\langle p, i \rangle}\| = 2\|w_p\| \quad ; \quad \left\| \frac{w_p}{\|w_p\|} - \frac{w_{\langle p, i \rangle}}{\|w_{\langle p, i \rangle}\|} \right\| = 2. \tag{11}$$

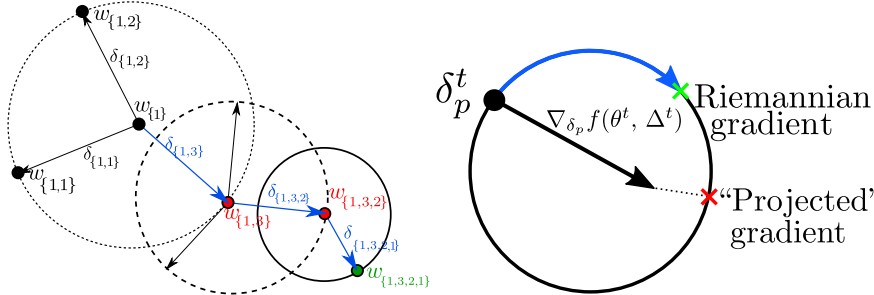

Figure 2: (**Left**) Example of hyper-planes $w_p$, formed through the sum of $\delta_p$. The hyper-plane $w_{\{1,3,2,1\}}$ associated to the class $n_{\{1,3,2,1\}}$ is in green, the construction with the $\delta$'s in blue, and all intermediate $w$ in red. (**Right**) Riemannian versus "projected" gradient descent. Riemannian optimization follows approximately geodesics, while projected gradient steps can jump very far from $\delta_p^t$.

This can be avoided, for example, by deploying the regularization parameter (or weight decay) independently for each $\|\delta_p\|$. However, it is costly in terms of hyper-parameter estimation.

In order to enforce the closeness of embedding vectors whose paths are similar, we penalize large norms of $\delta$. We also want to bound it away from zero to avoid the problem of separators that point in different direction may have a small norm. This naturally leads to a spherical constraint. Indeed, we transform the $\ell_2$ regularization over $\delta_p$ by fixing its norm in advance, i.e.,

$$\|\delta_p\| = R_p > 0. \tag{12}$$

In other words, we define $\delta_p$ on a sphere of radius $R_p$. The fully-connected layer $\Delta$ is then constrained on spheres, hence it is named *spherical fully-connected layer*.

Hence, we have $w_{\langle p,i\rangle}$ constrained on a sphere centered at $w_p$. This constraint prevents the direction of $w_{\langle p,i\rangle}$ from being too different from that of $w_p$, while bounding the distance away from zero. This does not add hyperparameters: instead of weight decay, we have the radius $R_p$ of the sphere.

### 3.1 RADIUS DECAY W.R.T. PATH LENGTH

We allow the radius of the spheres, $R_p$, to be defined as a function of the path. In this study, we use a simple strategy called *radius decay*, where $R_p$ decreases w.r.t. the path length:

$$R_p = R_0 \gamma^{|p|}, \tag{13}$$

where $R_0$ is the initial radius, $\gamma$ is the radius decay parameter, and $|p|$ is the length of the path. The optimal radius decay can be easily found using cross-validation. The radius decay is applied prior to learning (as opposed to weight-decay); then, the radius remains fixed during the optimization. As opposed to weight-decay, whose weight are multiplied by some constant smaller than one after each iteration, the radius decay here depends only on the path length, and the radius remains fixed during the optimization process.

The simplest way to apply the radius decay is by using the following predefined diagonal matrix $\mathbf{D}$,

$$\mathbf{D}_{i,i} = R_0 \gamma^{|p_i|}, \quad p_i \in \mathcal{P}, \quad 0 \text{ otherwise}, \tag{14}$$

where $p_i$ follows the ordering from Definition 1. Finally, the last layer of the neural network reads,

$$\underbrace{\phi_\theta(\cdot)}_{\text{Network}} \underbrace{\boxed{\Delta \mathbf{D} \mathbf{H}}}_{\text{Last layer}}. \tag{15}$$

The only learnable parameter in the last layer is $\Delta$.

### 3.2 OPTIMIZATION

There are several ways to optimize the network in the presence of the *spherical fully-connected layer*: by introducing the constraint in the model, "naively" by performing normalization after each step, or

by using Riemannian optimization algorithms. For simplicity, we consider the minimization problem,

$$\min_{\theta, \Delta} f(\theta, \Delta), \tag{16}$$

where $\theta$ are the parameters of the hidden layers, $\Delta$ the spherical fully-connected layer from equation 8, and $f$ the empirical expectation of the loss of the neural network. For clarity, we use noiseless gradients, but all results also apply to stochastic ones. The superscript $\cdot^t$ denotes the $t$-th iteration.

### 3.2.1 INTEGRATION OF THE CONSTRAINT IN THE MODEL

We present the simplest way to force the column of $\Delta$ to lie on a sphere, as this does not require a dedicated optimization algorithm. It is sufficient to normalize the column of $\Delta$ by their norm in the model. By introducing a dummy variable $\tilde{\Delta}$, which is the normalized version of $\Delta$, the last layer of the neural network equation 15 reads

$$\tilde{\Delta} = \left[ \dots, \frac{\delta_p}{\|\delta_p\|}, \dots \right], \quad \phi_\theta(\cdot)\tilde{\Delta}\mathbf{DH}. \tag{17}$$

Then, any standard optimization algorithm can be used for the learning phase. Technically, $\Delta$ is *not* constrained on a sphere, but the model will act as if $\Delta$ follows such constraint.

### 3.2.2 OPTIMIZATION OVER SPHERES: RIEMANNIAN (STOCHASTIC) GRADIENT DESCENT

The most direct way to optimize over a sphere is to normalize the columns of $\Delta$ by their norm after each iteration. However, this method has no convergence guarantee, and requires a modification in the optimization algorithm. Instead, we perform Riemannian gradient descent which we explain only briefly in this manuscript. We give the derivation of Riemannian gradient for spheres in Appendix B.

Riemannian gradient descent involves two steps: first, a projection to the tangent space, and then, a retraction to the manifold. The projection step computes the gradient of the function *on the manifold* (as opposed to the ambient space $\mathbb{R}^d$), such that its gradient is *tangent* to the sphere. Then, the retraction simply maps the new iterate to the sphere. With this two-step procedure, all directions pointing *outside* the manifold, (i.e., orthogonal to the manifold, thus irrelevant) are discarded by the projection. These two steps are summarized below,

$$s^t = (\delta_p^t)^T \nabla_{\delta_p} f(\theta^t, \Delta^t) \cdot \delta_p^t - \nabla_{\delta_p} f(\theta^t, \Delta^t), \qquad \delta_p^{t+1} = \frac{\delta_p^t + h^t s^t}{\|\delta_p^t + h^t s^t\|}, \tag{18}$$

where $s^t$ is the projection of the descent direction to the tangent space, and $\delta_p^{t+1}$ is the retraction of the gradient descent step with stepsize $h$. In our numerical experiments, we used the Geoopt optimizer (Kochurov et al., 2020), which implements Riemannian gradient descent on spheres.

## 4 NUMERICAL EXPERIMENTS

We experimented the proposed method using five publicly available datasets, namely CIFAR100 (Krizhevsky, 2009), Caltech-UCSD Birds 200 (CUB200) (Welinder et al., 2010), Stanford-Cars (Cars) (Krause et al., 2013), Stanford-dogs (Dogs) (Khosla et al., 2011), and Tiny-ImageNet (Tiny-ImNet) (Deng et al., 2009). CUB200, Cars, and Dogs datasets are used for fine-grained visual categorization (recognizing bird, dog bleeds, or car models), while CIFAR100 and Tiny-ImNet datasets are used for the classification of objects and animals. Unlike the datasets for object classification, the fine-grained visual categorization datasets show low inter-class variances. See Appendix C.2 and C.3 for more details about the dataset and their hierarchy, respectively.

### 4.1 DEEP NEURAL NETWORK MODELS AND TRAINING SETTING

We used the deep neural networks (*ResNet* (He et al., 2016) and *DenseNet* (Huang et al., 2017)). The input size of the datasets CUB200, Cars, Dogs, and Tiny-ImNet is $224 \times 224$, and $32 \times 32$ pixels for CIFAR100. Since the input-size of CIFAR100 does not fit to the original ResNet and DenseNet, we used a smaller kernel size (3 instead of 7) at the first convolutional layer and a smaller stride (1 instead of 2) at the first block.

**Remark: we do not use pretrained networks.** All networks are trained from scratch, i.e., we did not use pre-trained models. This is because most publicly available pre-trained models used ImageNet for training while Dogs and Tiny-ImNet are parts of ImageNet.

We used the stochastic gradient descent (SGD) over 300 epochs, with a mini-batch of 64 and a momentum parameter of 0.9 for training. The learning rate schedule is the same for all experiments, starting at 0.1, then decaying by a factor of 10 after 150, then 255 epochs. All tests are conducted using NVIDIA Tesla V100 GPU with the same random seed. Settings in more detail are provided in the supplementary material. We emphasize that we used the same parameters and learning rate schedule for *all scenarios*. Those parameters and schedule were optimized for SGD on plain networks, but are probably sub-optimal for our proposed methods.

## 4.2 RESULTS

Tables 1 and 2 show a comparison of the results obtained with several baseline methods and our methods. The first method, "*Plain*", is a plain network for subclass classification without hierarchical information. The second one, "*Multitask*" is simply the plain network with multitask (subclass and super-class classification) setting using the hierarchical information. The third one, "*Hierarchy*", uses our parametrization $W = \Delta \mathbf{H}$ with the hierarchical layer $\mathbf{H}$, but the columns of $\Delta$ are not constrained on spheres. Then, "*+Manifold*" means that $\Delta$ is restricted on a sphere using the normalization technique from Section 3.2.1. Finally, "*+Riemann*" means we used Riemannian optimization from Section 3.2.2. We show the experimental results on fine-grained visual classification (Table 1) and general object classification (Table 2).

Note that the *multitask* strategy in our experiment (and contrary to our regularization technique) does require an additional hyper-parameter that combines the two losses, because we train classifiers for super-classes and sub-classes simultaneously.

### 4.2.1 FINE-GRAINED CATEGORIZATION

As shown in Table 1, our proposed parameterization significantly improves the test accuracy over the baseline networks (ResNet-18/50, DenseNet-121/160). Even the simple hierarchical setting which uses the hierarchical layer only (without spheres) shows superior performance compared to the baseline networks. Integrating the manifolds with Riemannian SGD further improves the generalization performance.

Surprisingly, the plain network with deeper layers shows degraded performance. This can be attributed to overfitting which does not occur with our regularization technique, where larger networks show better performance, indicating the high efficiency of our approach.

Table 1: Test accuracy (%) for fine-grained classification. Radius decay is fixed at 0.5.

| Dataset | Architecture | Baseline | | Proposed parametrization | | |
| | | Plain | Multitask | Hierarchy | +Manifold | +Riemann |
|---|---|---|---|---|---|---|
| CUB200 | ResNet-18 | 54.88 | 53.99 | 58.28 | 60.42 | **60.98** |
| | ResNet-50 | 54.09 | 52.17 | 57.59 | 59.00 | **60.01** |
| | DenseNet-121 | 50.55 | 56.61 | 61.10 | 60.22 | **61.98** |
| | DenseNet-161 | 50.91 | 60.67 | 60.67 | 62.73 | **63.55** |
| Dogs | ResNet-18 | 59.17 | 59.88 | 60.30 | **61.83** | 61.36 |
| | ResNet-50 | 57.44 | 58.97 | 59.31 | 59.81 | **63.70** |
| | DenseNet-121 | 56.00 | 64.39 | 62.19 | 64.95 | **65.89** |
| | DenseNet-161 | 55.49 | 64.23 | 65.28 | 65.68 | **65.90** |
| Cars | ResNet-18 | 79.83 | 82.85 | **84.96** | 84.74 | 84.16 |
| | ResNet-50 | 82.86 | 82.86 | 83.34 | 84.51 | **84.65** |
| | DenseNet-121 | 79.78 | 85.39 | 85.97 | **86.00** | 85.54 |
| | DenseNet-161 | 79.85 | 85.79 | 86.23 | **86.90** | 85.76 |

Table 2: Test accuracy (%) for object classification. Radius decay is fixed at 0.5 and 0.9 for CIFAR100 and Tiny-Imnet, respectively.

| Dataset | Architecture | Baseline | | Proposed parametrization | | |
|---------|--------------|----------|----------|-----------|-----------|----------|
| | | Plain | Multitask | Hierarchy | +Manifold | +Riemann |
| CIFAR100 | ResNet-18 | 69.47 | 69.37 | 70.89 | 70.06 | **71.89** |
| | ResNet-50 | 71.04 | 71.74 | 73.75 | 73.76 | **73.97** |
| | DenseNet-121 | 74.50 | 75.62 | 76.38 | **76.52** | 76.28 |
| | DenseNet-161 | 75.30 | 76.57 | 77.01 | **77.01** | 76.64 |
| Tiny-ImNet | ResNet-18 | 64.70 | 64.81 | 64.33 | 64.74 | **65.13** |
| | ResNet-50 | 66.43 | 66.39 | 66.52 | **66.67** | 65.69 |
| | DenseNet-121 | 64.27 | 67.15 | 67.19 | **67.86** | 67.45 |
| | DenseNet-161 | 67.22 | 67.62 | 67.63 | **68.95** | 67.82 |

### 4.2.2 OBJECT CLASSIFICATION

We show test accuracy (%) of our proposed methods with different network models using CIFAR-100 and Tiny-ImNet, in Table 2. From the table, it can be seen that the proposed method has better accuracy than the baseline methods. Compared to the fine-grained classification datasets, the general object classification datasets have less similar classes on the same super-class. In these datasets, our method achieved relatively small gains.

A higher inter-class variance may explain the lower improvement compared to fine-grained categorization. Nevertheless, for Tiny-ImNet, e.g., ResNet-18 (11.28M parameters) with our parametrization achieves better classification performance than plain ResNet-50 (23.91M parameters). The same applies to DenseNet-121 and DenseNet-161. These results indicate that our regularization technique, *which does not introduce new parameters in the embedding layer*, can achieve a classification performance similar to that of more complex models.

### 4.3 RIEMANNIAN VS. PROJECTED SGD

Overall, Riemannian SGD showed slightly superior performance compared to projected SGD for fine-grained datasets, although, in most cases, the performance was similar. For instance, with the *Dogs* dataset on Resnet-50, Riemannian SGD shows a performance 4% higher than the projected SGD. For object classification, Riemannian SGD performs a bit more poorly. We suspect that, owing to the different radius decay parameters (0.5 in Table 1 and 0.9 in Table 2), the learning rate of Riemannian SGD should have been changed to a larger value.

## 5 CONCLUSION AND FUTURE WORK

We presented a simple regularization method for neural networks using a given hierarchical structure of the classes. The method involves of the reformulation of the fully connected layer of the neural network using the *hierarchical layer*. We further improved the technique using spherical constraints, transforming the last layer into a *spherical fully-connected layer*. Finally, we compared the optimization of the neural network using several strategies. The reformulation using the hierarchical layer $\Delta\mathbf{H}$ and the spherical constraint had a considerable impact on the generalization accuracy of the network. The Riemannian optimization had a lower overall impact, showing sometimes significant improvement and sometimes similar to its projected counterpart.

In this paper, we used the proposed regularization technique only on classical architectures. In the future, it would be interesting to use it on other architectures, e.g. Inception and SqueezeNet, for embedding, e.g. Poincaré, and other applications, e.g. Natural Language Processing (NLP). Moreover, in this paper, we used a given hierarchy mostly based on taxonomy designed by experts. This hierarchical structure, which is convenient for humans, may not be most convenient for classification algorithms. A self-supervised algorithm that learns classification *and* the hierarchy may be convenient because we do not need to access a hierarchy and lead to better results (because the structure will be more adapted to the task).

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

## A   EXAMPLE OF HIERARCHICAL STRUCTURE

Consider a dataset composed by the following labels: *cats*, *dogs*, *apple*, *orange*. These labels can be organized trough a hierarchical structure, with super-classes *animal* and *fruit*. In such case, the set $\mathcal{P}$ is composed by

$$\mathcal{P} = \Big\{ \{\text{fruit}\}, \{\text{animal}\}, \{\text{fruit, apple}\}, \{\text{fruit, orange}\}, \{\text{animal, cat}\}, \{\text{animal, dog}\} \Big\},$$

while the set $\mathcal{L}$ is composed by

$$\mathcal{L} = \Big\{ \{\text{fruit, apple}\}, \{\text{fruit, orange}\}, \{\text{animal, cat}\}, \{\text{animal, dog}\} \Big\}.$$

Then, its hierarchical layer reads (labels were added to ease the reading)

|  |  | {fruit, apple} | {fruit, orange} | {animal, cat} | {animal, dog} |
|---|---|---|---|---|---|
|  | {fruit} | **1** | **1** | 0 | 0 |
|  | {animal} | 0 | 0 | 1 | 1 |
| **H** = | {fruit, apple} | **1** | 0 | 0 | 0 |
|  | {fruit, orange} | 0 | **1** | 0 | 0 |
|  | {animal, cat} | 0 | 0 | 1 | 0 |
|  | {animal, dog} | 0 | 0 | 0 | 1 |

## B   OPTIMIZATION OVER SPHERES: RIEMANIAN (STOCHASTIC) GRADIENT DESCENT

We quickly recall some elements of optimization on manifolds, see e.g. Boumal (2020); Absil et al. (2007). For simplicity, we consider the optimization problem

$$\min_{x \in \mathcal{S}^{d-1}} f(x) \tag{19}$$

where $\mathbb{S}^{d-1}$ is the sphere manifold with radius one centered at zero and embedded in $\mathbb{R}^d$. The generic Riemannian gradient descent with stepsize $h$ reads

$$s_k = -\mathrm{grad} f(x_k) \tag{20}$$

$$x_{k+1} = R_{x_k}(h s_k) \tag{21}$$

where $\mathrm{grad} f$ is the gradient of $f$ *on the sphere*, which is a vector that belongs to the tangent space $\mathcal{T}_{x_k}\mathcal{S}^{d-1}$ (plane tangent to the sphere that contains $x_k$), and $R_{x_k}$ is a second-order retraction, i.e., a mapping from the tangent space $\mathcal{T}_{x_k}\mathcal{S}^{d-1}$ to the sphere $\mathcal{S}^{d-1}$ that satisfies some smoothness properties. The vector $s_k$ (that belongs to the tangent sphere) represents the local descent direction. We illustrate those quantities in Figure 3. Stochastic Riemannian gradient descent directly follows from (Bonnabel, 2013), replacing the gradient by its stochastic version.

In the special case of the sphere, we have an explicit formula for the tangent space and its projection, for the Riemannian gradient, and for the retraction:

$$T_x \mathbb{S}^{d-1} = \{y : y^T x = 0\} \quad ; \quad P_x(y) = y - (x^T y)x; \tag{22}$$

$$\mathrm{grad} f(x) = P_x(\nabla f(x)) \quad ; \quad R_x(y) = \frac{x+y}{\|x+y\|}. \tag{23}$$

The retraction is not necessarily unique, but this one satisfies all requirement to ensure good convergence properties. The gradient descent algorithm on a sphere thus reads

$$s_k = \left( (x_k^T \nabla f(x_k)) \, x_k - \nabla f(x_k) \right) \tag{24}$$

$$x_{k+1} = \frac{x_k + h_k s_k}{\|x_k + h_k s_k\|} \tag{25}$$

In our case, we have a matrix $\Delta$, whose each column $\delta_p$ belongs to a sphere. It suffices to apply the Riemannian gradient descent separately on each $\delta_p$. For practical reasons, we used the toolbox Geoopt (Kochurov et al., 2020; Bécigneul & Ganea, 2018) for numerical optimization.

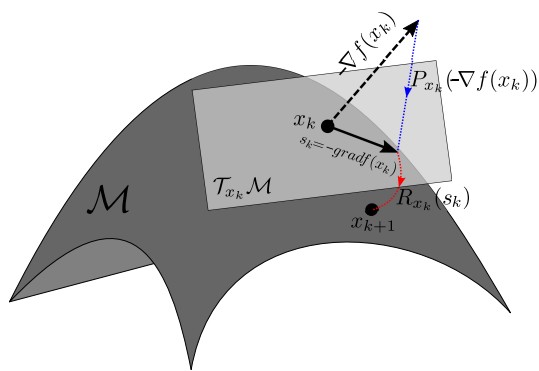

Figure 3: Illustration of a Riemanian gradient step on a manifold $\mathcal{M}$. In blue, the *projection* operator from the ambient space (in our case, $\mathbb{R}^d$) to the tangent space $T_{x_k}\mathcal{M}$. This maps the standard gradient of the function, $\nabla f$, to its Riemannian gradient $\mathrm{grad} f$. Then, in red we have the *retraction* that maps vector from the tangent space $T_{x_k}\mathcal{M}$ to the manifold $\mathcal{M}$. This converts the gradient step $x_k - h\mathrm{grad} f(x_k)$ (that belongs to the tangent space) to $x_{k+1}$ (that belongs to the manifold).

Table 3: Size of the datasets used in our experiments. $|L|$ denote the number of classes in the dataset, and $|P|$ the total number of classes and super-classes.

| Dataset | CIFAR100 | CUB200 | Stanford Cars | Stanford dogs | Tiny-ImNet |
|---|---|---|---|---|---|
| $|\mathcal{L}|$ | 100 | 200 | 196 | 121 | 200 |
| $|\mathcal{P}|$ | 120 | 270 | 205 | 194 | 295 |
| # Samples | 50k | 6k | 16k | 21k | 100k |

## C  NUMERICAL EXPERIMENTS: SUPPLEMENTARY MATERIALS

### C.1  DEEP NEURAL NETWORK MODELS AND TRAINING DETAILS

We used ResNet which consists of the basic blocks or the bottleneck blocks with output channels [64, 128, 256, 512] in Conv. layers. A dimensionality of an input vector to the FC layer is 512. We used DenseNet which includes hyperparameters such as ["growth rate", "block configuration", and "initial feature dimension"] for 'DenseNet-121' [32, (6, 12, 24, 16), 64] and 'DenseNet-161' [48, (6, 12, 36, 24), 96], respectively. A dimensionality of an input vector for DenseNets to the FC layer is 64 and 96.

Parameters in our proposed method using ResNet and DenseNet are optimized using the SGD with several settings: we fixed 1) the weight initialization with *Random-Seed* number '0' in pytorch, 2) learning rate schedule [0.1, 0.01, 0.001], 3) with momentum 0.9, 4) regularization: weight decay with 0.0001. A bias term in the FC layer is not used. The images (CUB200, Cars, Dogs, and Tiny-ImNet) in training and test sets are resized to $256 \times 256$ size. Then, the image is cropped with $224 \times 224$ size at random location in training and at center location in test. Horizontal flipping is applied in training. The learning rate decay by 0.1 at [150, 225] epochs from an initial value of 0.1. The experiments are conduced using GPU "NVIDIA TESLA V100". We used one GPU for ResNet-18, and two GPUs for ResNet-50, DenseNet-121, and DenseNet-161.

### C.2  DATASET

We summarize the important information of the previous datasets in Table 3. The next section describe how we build the hierarchical tree for each dataset.

### C.3  HIERARCHY FOR DATASETS

In this section, we describe how we build the hierarchy tree for each dataset. We provide also the files containing the hierarchy used in the experiments in the folder `Hierarchy_files`.

Before explaining how we generate the hierarchy, we quickly describe the content of the files. Their name follow the pattern `DATASETNAME_child_parent_pairs.txt`. The first line in the file corresponds to the number of entries. Then, the file is divided into two columns, representing pairs of (`child`, `parent`). This means if the pair $(n_1, n_2)$ exists in the file, the node $n_2$ is the direct parent of the node $n_1$. All labels have been converted into indexes.

### C.3.1 CIFAR100

The hierarchy of Cifar100 is given by the authors.

### C.3.2 CUB200

We classified the breed of birds into different groups, in function of the label name. For instance, the breeds `Black_footed_Albatross`, `Laysan_Albatross` and `Sooty_Albatross` are classified in the same super-class *Albatross*.

### C.3.3 STANFORD CARS

We manually classified the dataset into nine different super-classes: *SUV, Sedan, Coupe, Hatchback, Convertible, Wagon, Pickup, Van* and *Mini-Van*. In most cases, the super-class name appears in the name of the label.

### C.3.4 STANFORD DOGS

The hierarchy is recovered trough the breed presents at the end of the name of each dog specie. For instance, *English Setter, Irish Setter*, and *Gordon Setter* are classified under the class *Setter*.

### C.3.5 (TINY) IMAGENET

The labels of (tiny-)Imagenet are also Wordnet classes. We used the Wordnet hierarchy to build the ones of (Tiny) Imagenet. There are also two post-processing steps:

1. Wordnet hierarchy is not a tree, which means one node can have more than one ancestor. The choice was systematic: we arbitrarily chose as unique ancestor the first one in the sorted list.
2. In the case where a node has one and only one child, the node and its child are merged.

### C.4 GENERALIZATION PERFORMANCE ALONG DIFFERENT RADIUS DECAY VALUES

In this section, we show in Table 4 how the radius decay affects the test accuracy. In all experiments, we used the Resnet18 architecture with Riemannian gradient descent to optimize the spherical fully-connected layer.

Globally, we see that radius decay may influence the accuracy of the network. However, in most cases, the performance is not very sensitive to this parameter. The exception is for tiny-imagenet, where the hierarchy tree has many levels, and thus small values degrade a lot the accuracy.

Table 4: Influence of radius decay on the test performance for ResNet18.

| Radius Decay | CUB200 | Dogs | Cars | CIFAR100 | Tiny-ImageNet |
|---|---|---|---|---|---|
| 0.50 | 60.98 | 61.35 | 84.74 | 71.65 | 38.38 |
| 0.55 | 60.74 | 61.28 | 84.75 | 71.16 | 47.77 |
| 0.60 | 60.84 | 60.93 | 84.53 | 71.21 | 55.66 |
| 0.65 | 59.79 | 60.09 | 84.80 | 71.03 | 60.44 |
| 0.70 | 58.72 | 60.19 | 84.43 | 71.01 | 62.03 |
| 0.75 | 58.87 | 60.57 | 84.72 | 70.44 | 63.28 |
| 0.80 | 58.89 | 60.12 | 84.70 | 70.79 | 64.10 |
| 0.85 | 57.47 | 60.46 | 84.67 | 70.29 | 64.45 |
| 0.90 | 58.51 | 60.07 | 84.47 | 70.49 | 64.16 |
| 0.95 | 56.51 | 58.67 | 84.78 | 70.63 | 64.14 |
| 1.00 | 57.13 | 58.21 | 84.63 | 70.76 | 64.60 |

## C.5 LEARNING RADIUS DECAY

Here, we replace the diagonal matrix $\mathbf{D}$ in equation 14 with a learnable parameter matrix which is trained using backpropagation without an additional constraint or a loss function for simplicity. As shown in Table 5, this learnable radius is not effective the in terms of an classification performance compared to that the predefined radius decay.

Table 5: Test accuracy (%) with a learnable radius. Clearly, using learnable parameter degrades considerably the performance using the predefined radius decay (shown in parentheses, selected from Table 1).

| | | Proposed parametrization | | |
|---|---|---|---|---|
| Dataset | Architecture | Hierarchy | +Manifold | +Riemann |
| CUB200 | ResNet-18 | 53.40 (58.28) | 58.35 (60.42) | 58.24 (60.98) |
| Cars | ResNet-18 | 81.51 (84.96) | 82.54 (84.74) | 82.40 (84.16) |

## C.6 LEARNING WITH RANDOM HIERARCHY

As shown in Table 6, the methods with a randomly generated hierarchy showed a degraded performance compared to that with a reasonable hierarchical information.

Table 6: Test accuracy (%) with a random hierarchy tree. Radius decay is fixed at $0.5$. Clearly, using a random hierarchy degrades considerably the performance (shown in parentheses, selected from Table 1 which are the results with the original hierarchical tree). This validates the importance of the proper hierarchy information.

| | | Baseline | Proposed parametrization | | |
|---|---|---|---|---|---|
| Dataset | Architecture | Multitask | Hierarchy | +Manifold | +Riemann |
| CUB200 | ResNet-18 | 47.55 (53.99) | 50.28 (58.28) | 56.96 (60.42) | 56.43 (60.98) |
| Cars | ResNet-18 | 79.98 (82.85) | 81.07 (84.96) | 82.02 (84.74) | 81.84 (84.16) |

## C.7 SUPER-CLASS CLASSIFICATION EFFICIENCY

As shown in Table 7, our proposed methods (Hierarchy, +Manifold, and +Riemann) outperformed a multitask (multilabel) classification method in terms of test accuracy performance. Note that, in the multitask classification, a loss function for classification using superclasses is used additionally.

Table 7: Test accuracy (%) for **super-class** classification. Radius decay is fixed at $0.5$. For the super-class classification, without any modification of the proposed layers, we trained the model on the dataset, then calculate classification accuracy using the $\delta$'s corresponding to the parent classes.

| | | Baseline | Proposed parametrization | | |
|---|---|---|---|---|---|
| Dataset | Architecture | Multitask | Hierarchy | +Manifold | +Riemann |
| CUB200 | ResNet-18 | 53.68 | 58.87 | 61.17 | **62.22** |
| Cars | ResNet-18 | 86.88 | 87.91 | **91.23** | 90.97 |

## C.8 VISUALIZATION OF EMBEDDING VECTORS

In this section, we visualized an embedding vector which is an input of the last classification layer. As suggested by the reviewer, first, we show a distribution of two-dimensional vector ($\mathbb{R}^2$) learned by the networks. To obtain these vectors, we added new layers (i.e. a mapping function $\mathbb{R}^m \mapsto \mathbb{R}^2$, $m = 512$) prior to the last FC layer of ResNet-18. As two-dimensional vector is not enough to represent a discriminative feature for the fine-grained dataset which have a large number of classes different from MNIST dataset with ten classes with gray level images, we observe multidimensional vectors used in the ResNet. Second, we use t-SNE, to visualize the high dimensional embedding vector ($\mathbb{R}^{512}$) of the original ResNet, which is one of popular methods for exploring high-dimensional vector, introduced by van der Maaten and Hinton. Even though this method is known to have limitation which is highly dependent on hyperparameters such *perplexity* values, it is still useful to observe distribution of those high dimensional vectors by using fixed hyperparameters. For a deterministic way of visualization on 2D plane regardless of hyperparameters, finally, we visualized the embedding vectors using the traditional dimension reduction technique, namely Principal Component Analysis (PCA). Note that t-SNE and PCA have complementary characteristics (e.g., stochastic vs. deterministic, non-linear vs. linear, capturing local vs. global structures).

**1) Learned two-dimensional representation.** We show a distribution of two-dimensional embedding vector which is used for a classification. As shown in Figure 4, embedding vectors of our proposed methods are distributed more closely (clustered) with regard to their superclasses (Figure 4c,4d, and 4e). We used Cars dataset since the number of superclasses (nine super-classes) is small enough to show their distribution clearly. We show the results where the classification performances of all compared methods are similar. We captured the distribution of embedding vectors from an early epoch due to their a slow convergence rate. Two-dimensional vector seems too small to classify images of 196 classes which is highly non-linearly distributed.

**2) Observation of locality preserving structure via t-SNE.** We visualize high-dimension embedding vector extracted from ResNet-18 which is mapped into two-dimensional space by preserving local pairwise relationship of those vectors, using t-Distributed Stochastic Neighboring Entities (t-SNE). As in Figure 5, embedding vectors of our proposed methods are clearly clustered (Figure 5c,5d, and 5e) with regard to their superclasses compared to that of the baseline methods (Figure 5a and 5b).

**3) Observation of global structure via PCA.** Using PCA, we capture a global structure of high dimensional embedding vectors by projection onto two-dimensional space. As shown in Figure 6, embedding vectors extracted from ResNet-18 of our proposed methods have less-Gaussian shape distribution (Figure 6d and 6e) with regard to their superclasses than that of the baseline methods (Figure 6a and 5b). We observe that individual distribution of each classes is similar to that of Figure 4.

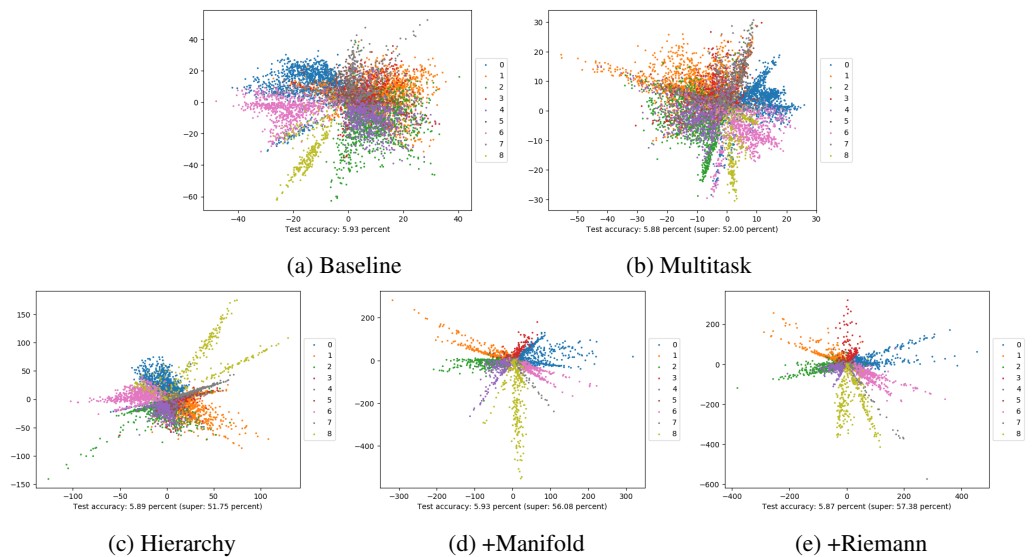

Figure 4: **Visualization of 2-dimensional embedding vector.** We added the fully connected layer ($\mathbb{R}^{m \times 2}$, e.g., for ResNet18, $m = 512$) prior to the last FC layer. **(a) Baseline**, **(b) Baseline (Multitask)**, and proposed parameterization ((**c) Hierarchy**, **(d) +Manifold**, and **(e) +Riemann**).

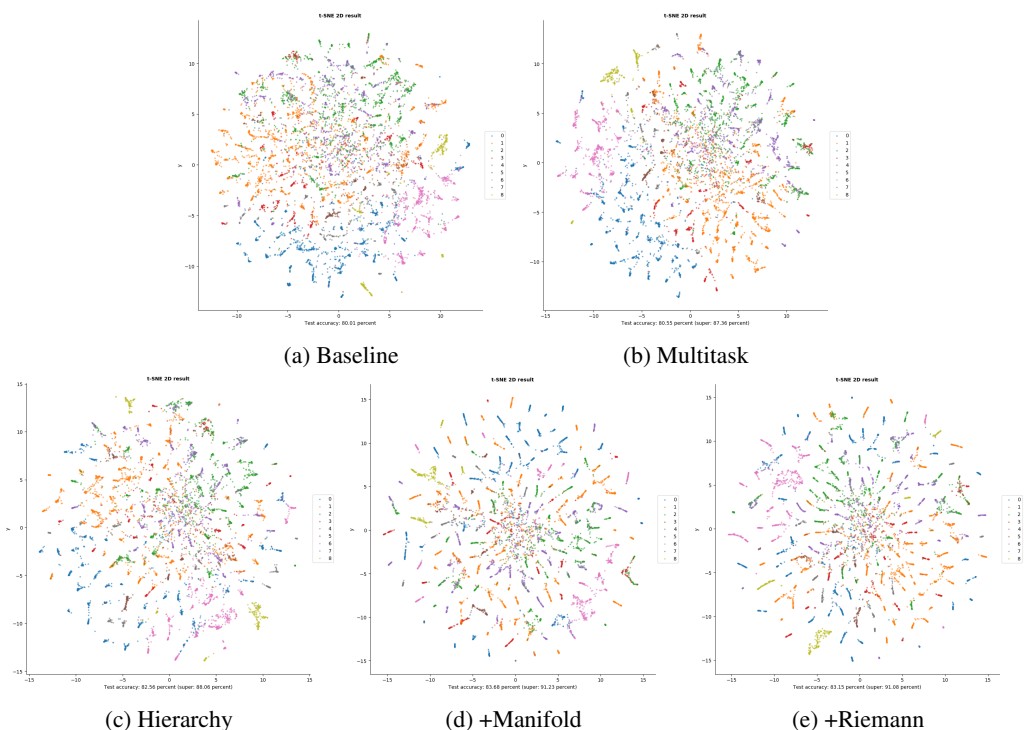

Figure 5: **Visualization of a high dimensional embedding vector using t-SNE on 2D plane. (a) Baseline**, **(b) Baseline (Multitask)**, and proposed parameterization ((**c) Hierarchy**, **(d) +Manifold**, and **(e) +Riemann**).

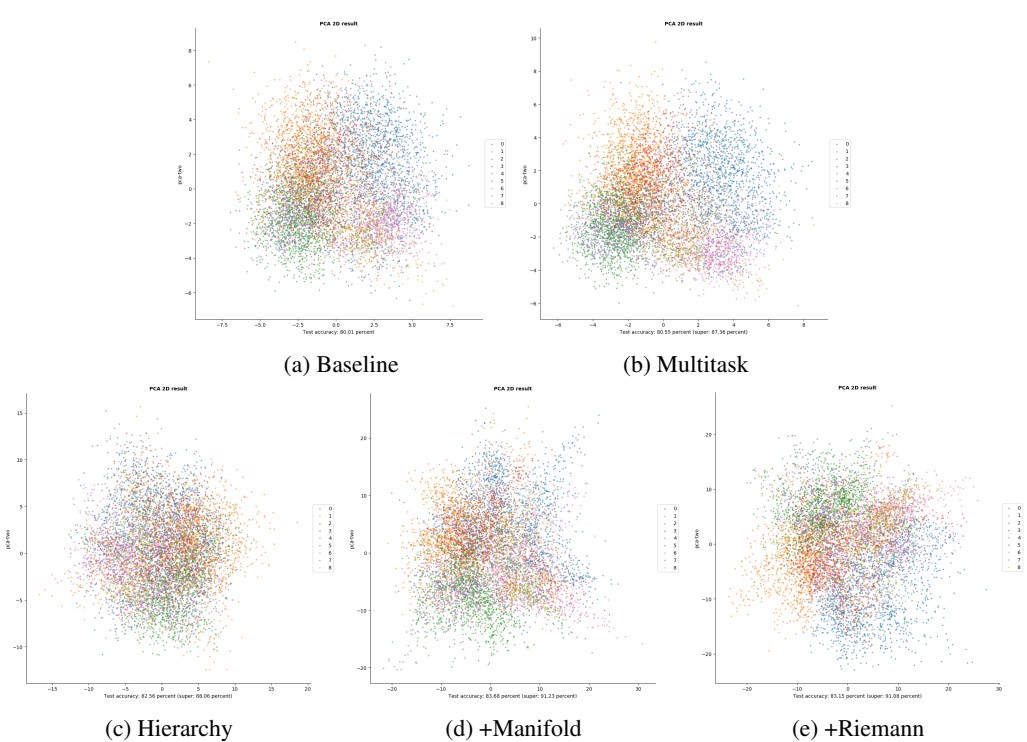

(a) Baseline         (b) Multitask

(c) Hierarchy       (d) +Manifold       (e) +Riemann

Figure 6: **Visualization of a high dimensional embedding vector on 2D plane using PCA. (a) Baseline (b) Baseline (Multitask)**, and proposed parameterization (**(c) Hierarchy**, **(d) +Manifold**, and **(e) +Riemann**).

