# OpenReview forum: "Connecting Sphere Manifolds Hierarchically for Regularization"
_ICLR.cc/2021/Conference — Reject_

### Official Review · AnonReviewer1 · 2020-10-26
**misleading statements and justifications should be addressed**

**Rating:** 5
**Confidence:** 3

**Review:**

Section 1: The third paragraph is confusing. It is not clear why Euclidean distance is not sufficient for learning with such a hierarchical regularization. Can't the parent class mean just the mean of all its children?

Section 1: From the second last paragraph, it reads that the paper does not train the neural network, but adopts a pre-trained off-the-shelf network and works on its last layer. While this makes the method simple and compatible to other network architectures, is there a reason not to train an end-to-end network with the proposed technique? However, in Section 4.1, the paper says all networks are trained from scratch. Does the training of the whole model follow end-to-end training or stage-wise training?

Section 2.2: How to define "separator"? "Separator" does not seem like a formal word. Moreover, the sentence is confusing, "the parameter of the classifiers that identify dog’s breed should also be similar". If similar, how can they differentiate dog breeds?

Section 3.1: Is radius decay a new method proposed in the paper? If so, what is the rationale behind this design? If not, are there any related work adopting this method? Why not use other linear decay methods, e.g., $R_p=R_0*|p|$?

Eq.(16) implies that the whole model (backbone and the proposed spherical fully-connected layer) is end-to-end optimized. What is the optimization method used for learning other layers?

Moreover, the paper says "the most direct way to optimize over a sphere is to normalize the columns of ∆ by their norm after each iteration. However, this method has no convergence guarantee, and requires a modification in the optimization algorithm". To construct the spherical fully-connected layer, isn't it equivalent to learn a normal fully-connected layer followed up by L2-normalization and a scaling operation?

Even though the authors provide C.2, C.3 and plain files on the prepared hierarchies in the datasets, it is disappointing that the paper does not present the hierarchical structure in a nice way. Perhaps a visualization on the class labels w.r.t the hierarchy serves the paper better?

Section 4.1: Tiny-ImageNet dataset seems to have lower resolution of images (64x64). When authors say "input size...224x224", do the authors mean that Tiny-ImageNet images are resized from 64x64 to 224x224?

Section 4.1: The paper trains all networks from scratch by explaining "Dogs and Tiny-ImNet are parts of ImageNet". Does it mean that images from datasets Dogs and Tiny-Imagenet are part of ImageNet? Or Does it mean that classes in the two datasets are included in the set of ImageNet classes?

Section 4.1: How to define "plain networks"? The paper uses two networks ResNet and DenseNet, then what is "plain networks"? Once the authors state that the parameters are "probably sub-optimal for our proposed methods", it also implies that the parameters may be even more sub-optimal to the compared methods?

Section 4.2.1: When the paper claims "high efficiency of our approach", it does not justify the "efficiency part". How to tell if the training or inference efficiency is higher than other methods?

Figure 2 right depicts Riemannian gradient and "projected gradient", but the paper does not formally compare them. It is not clear which one may be better than the other? For learning, gradient guides the direction to update parameters, but the scale in the update also matters. Is there a discussion on which one may be more efficient (or compute time) during a training iteration? Section 4.2 explicitly notes the performance difference between the two methods. The paper should discuss this further for better understanding. Moreover, given that the two methods are so different, setting the same learning rate schedule (cf. Section 4.1) is not sensible, because they can perform quite differently with different learning rates.

The paper does not discuss how the proposed method may work if classes do not follow a tree hierarchy, although C.3 talks a bit in the context of Tiny-ImageNet. As the paper focuses on a generic regularization based on hierarchical information, it may also need to discuss how this can be applied in multi-label classification problems.


---------------------------------
post-rebuttal
---------------------------------
I appreciate that authors have provided rebuttal that addresses many of my questions, though I'd like to maintain my initial rating due to the following comments. I think this paper is at the borderline.

In terms of explaining why "Euclidean distance is not sufficient for learning such a hierarchical regularization", I don't find the illustration example in Section 2.3 intuitive or concrete. I don't think Eq3 adds much as the paper does not explain further. Perhaps the confusion is from that the paper does not explicitly explain what "optimal classifier" mean in terms of Eq3.

The authors only say "those parameters and schedule were optimized for SGD on plain networks, probably sub-optimal for our proposed methods.” It is not clear whether other methods suffer severely from the choice of learning rate and scheduler. As far as I know, SGD is sensitive to the initial learning rate. So I am worried that setting the same learning rate is not fair to comparing different models that have different structures.

From the updated paper, I find the blue line in Page-2 confusing. It is not clear about the logic: why diversity reduces over-fitting. (Xie et al. 2017) studies this point with a complete paper. But the way that authors simply put it is quite unclear how this statement is related in the context.

Visualization is interesting to look at. But it should be better analyzed. For example, visually all methods produce similar tSNE visuals in Figure 5. But are there any essential difference?

---

> ### Author Response · Authors · 2020-11-22
> **Responses to AnonReviewer1 [R1] (Part 1)**
>
> We thank the reviewer for his detailed and careful review.
>
> > The third paragraph is confusing. It is not clear why Euclidean distance is not sufficient for learning with such a hierarchical regularization.
>
> We illustrate an example of why Euclidean distance may not be representative of the distance between two classifiers in section 2.3.
>
> > From the second last paragraph, it reads that the paper does not train the neural network but adopts a pre-trained off-the-shelf network and works on its last layer. [...] However, in Section 4.1, the paper says all networks are trained from scratch. [...]
>
> We applied end-to-end training without the pre-trained models. The meaning of “we do not change prior layers of the deep neural network” in the second last paragraph is that we do not change the architecture of the network except the last layer. We will make this statement more clear in the revised version
>
> > How to define "separator"? Moreover,
>
> We will revise “separator” with “hyperplane.”
>
> > the sentence is confusing, "the parameter of the classifiers that identify the dog’s breed should also be similar.” If similar, how can they differentiate dog breeds?
>
> The hypothesis we make in the paper is that two dogs with breeds from the same family (i.e. super-class) should share more common characteristics than two dogs from different classes. Hence, since they share similar characteristics, the two separating hyperplanes should point roughly in the same direction, except for a few features that allow their distinction.
>
> > Is radius decay a new method proposed in the paper? If so, what is the rationale behind this design? [...] Why not use other linear decay methods?
>
> The rationale behind the radius decay is similar to a “hard” l2 regularisation. We tried, as suggested  As pointed by R.3, “labels in finer-grained level should be modeled with less capacity.”
>
> A linear shrinking can be applied too. The major reason we choose the geometrical decrease is due to the main reason that we wanted the maximum radius to be bounded even for deep hierarchy.
>
> > What is the optimization method used for learning other layers?
>
> We used SGD in Euclidean space for the whole model except the proposed last layer. The parameters for SGD are the same as the one used for the original network.
>
> > To construct the spherical fully-connected layer, isn't it equivalent to learn a normal fully-connected layer followed up by L2-normalization and a scaling operation?
>
> Indeed, as suggested by the reviewer, it is possible to include normalization and scaling layers to the network to avoid the usage of another optimization algorithm (in the case where we use SGD). However, this does not apply to Riemannian gradient descent.
>
> > It is disappointing that the paper does not present the hierarchical structure in a nice way. Perhaps a visualization on the class labels w.r.t the hierarchy serves the paper better?
>
> We will work on the visualization of the hierarchy for the revised version.

---

> ### Author Response · Authors · 2020-11-22
> **Responses to AnonReviewer1 [R1] (Part 2)**
>
> > When authors say "input size...224x224", do the authors mean that Tiny-ImageNet images are resized from 64x64 to 224x224?
>
> We used the Tiny-ImageNet dataset with a 224x224 resolution image which is cropped from 256x256 (resized from 64x64 first). We will revise it.
>
> > The paper trains all networks from scratch by explaining "Dogs and Tiny-ImNet are parts of ImageNet". Does it mean that images from datasets Dogs and Tiny-Imagenet are part of ImageNet?
>
> As stated in Stanford Dogs, their images and annotation from ImageNet. And Tiny-ImageNet uses images from ImageNet. Therefore, we cannot use pre-trained networks, as they are usually trained on ImageNet.
>
> > How to define "plain networks"?
>
> By “plain network,” we meant we used the network without any modification, i.e., we used the same original architecture, initialization, hyper-parameter, and optimizer.
>
> > Once the authors state that the parameters are "probably sub-optimal for our proposed methods,” it also implies that the parameters may be even more sub-optimal to the compared methods?
>
> This means that we kept most hyperparameters unchanged, for instance, the learning rate schedule or the initialization. Those hyperparameters, which come along with a specific architecture, are usually optimized for this architecture, but not to our. This means that there is a potential improvement if we optimize over the other hyperparameters.
>
> > When the paper claims "high efficiency of our approach,” it does not justify the "efficiency part.” How to tell if the training or inference efficiency is higher than other methods?
>
> We meant “efficiency” in terms of the depth of the networks. The performance of our proposed method with shallower layers show a better generalization performance compared to that of deeper baseline networks.
>
> > Figure 2 right depicts the Riemannian gradient and "projected gradient,” but the paper does not formally compare them. It is not clear which one may be better than the other? [...] Moreover, given that the two methods are so different, setting the same learning rate schedule (cf. Section 4.1) is not sensible because they can perform quite differently with different learning rates.
>
> While we provide some more technical detail on the Riemannian gradient in Appendix B, we provide a comparison in detail between Riemannian (in Section 3.2.2.) and Projected gradient (Section 3.2.1.). Technically, Riemannian gradient descent has more desirable theoretical properties than gradient descent - but those results do not really apply to neural networks, as they are highly non-convex and non-smooth. We did not perform an extensive comparison between projected and Riemannian gradient descent as this is out of the scope of the paper, but our experiments suggest that Riemannian gradient descent tends to perform slightly better than its projected counterpart.
>
> > The paper does not discuss how the proposed method may work if classes do not follow a tree hierarchy, although C.3 talks a bit in the context of Tiny-ImageNet. As the paper focuses on a generic regularization based on hierarchical information, it may also need to discuss how this can be applied in multi-label classification problems.
>
> A possible way to improve the method to non-mutually exclusive hierarchy would be to consider the adjacency matrix of a graph of hierarchy rather than a tree. We plan to investigate this direction in future work.

---

### Official Review · AnonReviewer4 · 2020-10-28
**This paper proposes a technique to classify hierarchically organized classes.**

**Rating:** 5
**Confidence:** 4

**Review:**

The idea of introducing class hierarchy as a regularization into deep networks seems to be novel.

The following comments could be relevant:

1. The reviewer finds Section 2 is not easy to follow:
- Authors may consider to give more specific definitions to terms such as, classifier, separators, etc. For example, in (2), Wp and Wpi are called classifiers. What are they, hyperplanes?
- In Definition 1, authors may like to give some early examples about P and L. Otherwise, it is not easy to interpret the matrix H.
- Authors may consider to use a different notation for Delta in (8), as Delta may remind an operator on H in (9).

2. In (9), do we require or observe deltas in the same subtree roughly the same direction?

3. In Section 3, it is claimed no hyperparameters are added. However, it seems that, initial radius R0, radius decay parameter, even how to organize classes may all be considered as additional hyperparameters.

4. In reality, it can be non-trivial, or even impossible,  to define mutual exclusive class partitions to form the required class tree in Figure 1. Authors may discuss how different class hierarchy adopted affects the classification accuracy, e.g., in Table 2.

---

> ### Author Response · Authors · 2020-11-22
> **Responses to AnonReviewer4 [R4]**
>
> We thank the reviewer for his careful review and insightful comments.
>
> > 1. The reviewer finds Section 2 is not easy to follow:
> > Authors may consider giving more specific definitions to terms such as classifiers, separators, etc. For example, in (2), Wp and > Wpi are called classifiers. What are they, hyperplanes?
> > In Definition 1, authors may like to give some early examples of P and L. Otherwise, it is not easy to interpret the matrix H.
> > The authors may consider using a different notation for Delta in (8), as Delta may remind an operator on H in (9).
>
> 1. Separator means a classifier, which is a form of hyperplanes in the vector space. We will use one term “hyperplane” consistently and introduce a proper definition in the paper.
> 2. Meanwhile, we already defined the sets P and L in Section 2.1 prior to Definition 1. There is also an example of H, P, and L in Appendix A - Example of hierarchical structure.
> 3. We understand that the notation Delta may be confusing - we will make this clearer in the revision.
>
> > 2. In (9), do we require or observe deltas in the same subtree roughly the same direction?
>
> Not especially. In fact, we expect to observe that the delta’s to point toward different directions, as they represent the differences between the classifier of a class with the classifier of its superclass. This links to a remark made by R. 3 about diversity, where we can expect to have better performance by maximizing the distance between the delta of the same superclass.
>
> > 3. In Section 3, it is claimed no hyperparameters are added. However, it seems that the initial radius R0, radius decay parameter, even how to organize classes may all be considered as additional hyperparameters.
>
> The initial radius R0 does not play a role, as this scale the entire output by a constant. However, the radius decay is (as the reviewer mentioned) a hyperparameter, which can be determined by simple cross-validation.
>
> We understand that our statement is misleading, and this will be corrected in the revised version.
>
> > 4. In reality, it can be non-trivial, or even impossible, to define mutual exclusive class partitions to form the required class tree in Figure 1. The authors may discuss how different class hierarchy adopted affects classification accuracy, e.g., in Table 2.
>
> This is a critical point that is directly connected to explain why the generalization performance of object classification using mutual exclusive classes in Table 2 is not compared to that of fine-grained classification in Table 1. As [R3] also mentioned the importance of the hierarchy definition, we are going to observe the generalization performance along with different hierarchy settings.

---

### Official Review · AnonReviewer3 · 2020-10-31
**Interesting idea with some concerns**

**Rating:** 6
**Confidence:** 4

**Review:**

I generally like the neat idea of introducing hierarchical spheres to model the intra- and inter-class relationships among the hierachical labels. It is naturally motivated to combine the hierarchical structure in the label space as a prior knowledge to supervise the training of neural networks. The overall idea is simple and easy to understand. The method models the labels at different levels by adding a free vector that is constrained on a specific hypersphere, then uses a smart way of formulating this procedure with simple matrix multiplication, and finally considers an alternative manifold optimization method to train the neural network in an end-to-end fashion. As far as I'm concerned, the intuition has also been explored in [Deep neural decision forests, ICCV 2015], but differently, the ICCV 2015 paper considered the hierarchical label structure with a decision forest. I believe this direction is of sufficient significance to the ML community.

This paper has several aspects that I found most interesting:

(1) The formulation is interesting and is novel from my perspective. Modeling the fine-grained classes by successively adding a "perturbation" vector makes sense to me. Then, the authors are able to formulate this in a matrix multiplication, which is basically a linear matrix factorization that over-parameterizes the classifiers. Although technically the linear matrix multiplication is still equivalent to a linear classifier, the fact that it can still improve the network generalization is interesting and is partially verified by a number of theory works. Besides, as a way to combine prior knowledge to supervise the neural networks, such a simple linear matrix factorization (with some constraints like sphericity, radius decay, etc.) provides a potentially useful way to incorporate some regularization priors.

(2) The use of spherical constraints is interesting and empirically make senses to me. By constraining the learning on the spherical space can ease the training difficulties of the over-parameterized classification layers. This is, in fact, also observed and verified by [Neural Similarity Learning, Neurips 2019]. It will be potentially interesting to connect these two papers and have some discussions. The radius decay for the hierarchical spheres is also novel to me, because labels in finer-grained level should be modeled with less capacity.

Desipte these interesting aspects, I also have a few concerns and suggestions to improve the paper:

(1) The empirical evaluation is relatively weak and the evaluation metric seems not to well reflect the advantages of hierarchically modelling the label space. For example, I think it will be more informative to incorporate the classification accuracy of the super-classes. It will make this paper more interesting to have more experiments that analyzes the difference in feature distributions between normally trained neural networks and the hierarchically trained neural networks. For example, an intuitive visualization of the feature space will be of great interest. An easy way for the visualziaiton is to set the outpute feature dimension as 2 and directly plot them, similar to [A Discriminative Feature Learning Approach for Deep Face Recognition, ECCV 2016] and [Large-Margin Softmax Loss for Convolutional Neural Networks, ICML 2016].

(2) Some important ablation studies to justify some heuristic designs are very important and necessary. For example, there is a hyperprameter in the radius decay, how it will affect the performance is crucial. Potentially, the authors can also evaluate what if no sphericity constraint is applied, or what if no radius decay is used, etc. Since this paper proposes a number of heuristic designs, it is very important to justify them (either from theoretical perspective, or from empirical evaluations).

(3) Although I believe it is useful to model the hierachical label space in an explicit way, the empirical evaluation does not really convince me on that, especially experiments on CIFAR-100 and Tiny-ImageNet. The method uses additional prior knowledge on the label space, but only yields very limited performance gain. I think using some other SOTA regularization can easily improve more. What is the underlying reason? I think more discussions and insights will be useful.

(4) The usefulness of the hierachical label structure should be evaluated and verified in the first place. A simple way to evaluate it is to use some random assignment or simple K-means assignments for the super-classes. If using the ground truth hierachical strucutre can consistently outperform the random or K-means super-class assignment, then one can believe that incorporating the ground truth hierachical label structure is indeed useful. Until then, it makes little sense to argue it is beneficial to generalization to combine the ground truth hierachical label structure. I highly suggest the authors conduct such an experiment.

Some minor concerns and suggestions:

(5) I cannot find the Spherical CNN from Xie et al. (2017) on page 1. I think the paper is more closely related to [Deep Hyperspherical Learning, Neurips 2017] in terms of the spherical regularization. The authors may discuss the connections and differences to this paper.

(6) Since the authors consider regularizations for the intra-class hierachical label structure, it will be interesting to see whether the regularization on the inter-class regularization will be beneficial or not. For example, the authors can use some diversity regularization on sphere to push away classifiers from different super classes. A potential regularization for this is [
Learning towards Minimum Hyperspherical Energy, Neurips 2018]. I want to note that it is a suggestion for the paper rather than a weakness.

To summarize, I think the paper proposes a very interesting and potentially widely useful method to incorporate the hierachical label structure to train neural networks. Currently, I feel postive to accept this paper, and I am sitting between 6 and 7 (I give a 6 for now). I will consider to increase my score if the authors well address the concerns.

---

> ### Author Response · Authors · 2020-11-22
> **Responses to AnonReviewer3 [R3]**
>
> We thank the reviewer for its positive review. We will clarify the concerns raised by the reviewer.
>
> > (1) The empirical evaluation is relatively weak, and the evaluation metric seems not to well reflect the advantages of hierarchically modeling the label space. For example, I think it will be more informative to incorporate the classification accuracy of the super-classes. [...] An intuitive visualization of the feature space will be of great interest. [...]
>
> Classification accuracy of the super-classes and visualization of embedding vectors: we appreciate the reviewer’s constructive suggestion. We are going to observe it in the revision. Please see experimental results shortly and C.7 and Table 7 for classification accuracy of superclasses and C.8 and Figure 4 for visualization in the revised manuscript.
>
> > (2) Some important ablation studies to justify some heuristic designs are very important and necessary. For example, there is a hyperparameter in the radius decay. How it will affect the performance is crucial. Potentially, the authors can also evaluate what if no sphericity constraint is applied, or what if no radius decay is used, etc. Since this paper proposes a number of heuristic designs, it is very important to justify them (either from a theoretical perspective or from empirical evaluations).
>
> Radius could be a learnable parameter that can be optimized. We have an experiment along with different radius decay in the appendix (Table 4.) Also, the third column of table 1 and 2 corresponds to the no sphericity constraint (i.e., only the hierarchical layer is used). Finally, if we do not use the radius decay, we refer the reviewer to the last line of Table  4, and we clearly see that the performance of the network is greatly affected. Please see experimental results above and C.5 and Table 5 in the revised manuscript in detail.
>
> > (3) Although I believe it is useful to model the hierarchical label space in an explicit way, the empirical evaluation does not really convince me on that, especially experiments on CIFAR-100 and Tiny-ImageNet. The method uses additional prior knowledge on the label space but only yields very limited performance gain. I think using some other SOTA regularization can easily improve more. What is the underlying reason? I think more discussions and insights will be useful.
>
> As described in the manuscript, general object classification datasets have less similar classes compared to the fine-grained ones. This partially breaks our hypothesis that elements from similar classes should share similar features. One other reason may be that using semantic hierarchy does not reflect the similarity between classes in the Imagenet dataset.
>
> > (4) The usefulness of the hierarchical label structure should be evaluated and verified in the first place. A simple way to evaluate it is to use some random assignment or simple K-means assignments for the super-classes. [...] I highly suggest the authors conduct such an experiment.
>
> We agree that a definition of the hierarchy is one of the important factors to have a better generalization performance. We appreciate the reviewer’s interesting study suggestion. We also do believe that combining a method that models the hierarchy based on inter-class similarities will substantially improve the performance of our method. Please see experimental results above and C.6 and Table 6 in the revised manuscript in detail.
>
> > (5) I cannot find the Spherical CNN from Xie et al. (2017) on page 1 [...]
>
> We added it to the revision (in introduction).
>
> > (6) Since the authors consider regularizations for the intra-class hierarchical label structure, it will be interesting to see whether the regularization on the inter-class regularization will be beneficial or not. For example, the authors can use some diversity regularization on the sphere to push away classifiers from different superclasses.
>
> Since we applied regularization along with the depth of hierarchy, we applied regularization to both inter-class and intra-class. The idea of diversity is interesting - we actually have envisaged exploring this direction - but we finally decided to focus more on our contribution instead, as diversity was out-of-the-scope of our study,

---

### Official Review · AnonReviewer2 · 2020-11-03
**Interesting method but rather weak experiment**

**Rating:** 5
**Confidence:** 4

**Review:**

In this paper, the authors proposed a novel reparameterization framework of the last network layer that takes semantic hierarchy into account. Specifically, the authors assume a predefined hierarchy graph, and model the classifier of child classes as a parent classifier plus offsets $\delta$ recursively. The authors show that such hierarchy can be parameterized a matrix multiplication $\Delta \mathbf{H}$ where $\mathbf{H}$ is predefined by the graph. In addition, the authors further propose to fix the norm of $\delta$ in a decaying manner with respect to path length. The resulting spherical objective is optimized via Riemannian gradient descent.

The strengths and weaknesses are very obvious in this paper.

On the strength side:
+ The paper itself is very well written. The notations are well defined and the methods are very clearly explained. The presentation is fluent.
+ The proposed method seems novel and interesting. The derivations are technically correct.
+ Experiments show performance improvement over baselines, especially on CUB200/Dogs/Cars.

On the weakness side:
- I think experiment presents the most significant weakness of this paper: 1) The comparison is rather weak without any reference to existing prior arts such as [1]. A simple search with respect to the 5 experiment datasets also show significant performance gaps between the proposed method and latest methods. 2) I remain skeptical about the solidness of the baseline performance as they show considerable gaps to standard baseline training without bells and whistles (https://github.com/weiaicunzai/pytorch-cifar100). 3) The performance gain diminishes very quickly on bigger dataset such as Tiny ImageNet. What about the results on ImageNet?
- The proposed method depends on a pre-defined semantic hierarchical graph rather than a learned one, which potentially limits the technical value of this work. In certain cases, semantic hierarchy may not always be a reasonable choice to guide the learning of visual embedding.
- I have some concern about the selection of initial radius $R_0$ and its decay policy. I think this parameter should be dataset dependent due to different numbers of categories and the densities of class distributions. As a result, how such parameter and policy can be optimally determined becomes a question.
- Finally, forcing a fixed radius does not sound as reasonable as allowing a learnable radius with soft regularization.

[1] Chen et al., Fine-grained representation learning and recognition by exploiting hierarchical semantic embedding, ACM-MM 2018

========================== Post Rebuttal ==============================

The authors did a good job in addressing some of my concerns in the rebuttal. Thus I am increasing the score in response to the clarifications. However, I feel there is still some improvement space for the experiment part of this section, and I encourage the authors to incorporate the changes, including ImageNet experiment and following stronger baselines to make the results more solid and convincing.

---

> ### Author Response · Authors · 2020-11-22
> **Responses to AnonReviewer2 [R2]**
>
> We thank the reviewer for his careful review and insightful comments. It seems that the major concerns are 1) the experiments, 2) the hyperparameters and 3) the non-learnable hierarchy. We will answer all concerns below. Regarding 1) and 2), a response can be found in the common answer above (https://openreview.net/forum?id=hbzCPZEIUU&noteId=mS9kbsYdJVA).
> We hope that our responses are satisfying the reviewer's questions.
>
> > On the weakness side:
> > I think the experiment presents the most significant weakness of this paper: 1) The comparison is rather weak without any reference to existing prior arts such as [1]. [...] 2) I remain skeptical about the solidness of the baseline performance as they show considerable gaps to the standard baseline [...]. 3) The performance gain diminishes very quickly on bigger datasets such as Tiny ImageNet.
>
> 1) Prior arts HSE in [1], the authors use an input resolution of 448x448 with the pre-trained model for their experiments (https://github.com/tianshuichen/HSE/tree/master/code/CUB_200_2011/HSE), which is one factor that explains their high accuracy on the baseline. Furthermore, in [1], an input image is used without cropping the object (i.e., bird) which might involve background rather than the bird itself. The improvement (max 12.6%, min 5.92%) using our proposed methods shows more than that of the HSE method [1] (2.9%), compared to baseline, respectively.
>
>
> |  		       order    |   family   |  genus |    class|
> | ------------- |:-------------:| -----:| -----:|
> |baseline       |	98.8	|	95.0	 |	91.5	|	85.2|
> |HSE(ours) | 	98.8	|	95.7	|	92.7	|	88.1|
> Table: performance on [1].
>
> 2) In https://github.com/weiaicunzai/pytorch-cifar100, there are additional ways, which are not used in our experiments, such as the warm-up epoch, which uses a bigger learning rate during the first few epochs can improve the learning convergence, and the mean and standard variation calculated using CIFAR100 gives a better generalization accuracy.
>
> 3) The performance diminishes on object classification, which hierarchy could not be mutually exclusive, but this is not due to the number of samples. We observed that the performance gain on ImageNet is similar to TinyImageNet. We will add it in the revision.
>
> > The proposed method depends on a pre-defined semantic hierarchical graph rather than a learned one
>
> We suppose we can fix the problem of mutually exclusive hierarchies by considering a graph rather than a tree. We do think this could improve the performance of our approach on datasets like ImageNet. We left this part for future work.
>
> We also agree with the reviewer that the fixed hierarchy is a limiting factor in our approach. As seen in the experiments, the semantic hierarchy is definitively not the best to classify objects in datasets such as ImageNet. However, in the case where the hierarchy is based on features, such as a dog’s breed, there is a clear, big gap in the performance between standard and modified networks. We also suppose it may be possible to combine methods that learn the hierarchy with our that exploit the hierarchical structure.
>
> Combining our approach with dynamic hierarchy is definitely a very interesting way to explore, but this is out-of-the-scope of the paper and this is left for future work.
>
> > I have some concerns about the selection of initial radius R0 and its decay policy. I think this parameter should be dataset dependent due to different numbers of categories and the densities of class distributions. As a result, how such parameters and policies can be optimally determined becomes a question.
>
> - The initial radius does not play a role, as this scales the entire output. This is why we fixed it to R0 = 1.
> - The radius decay, however, is a learnable parameter that can be optimized. We added an experiment along different radius (decay) in the appendix (Table 4.)
>
> > Finally, forcing a fixed radius does not sound as reasonable as allowing a learnable radius with soft regularization.
>
> We agree with the reviewer that a learnable radius is an interesting direction. We will try a learnable radius for further improved performance. However, forcing the radius may also help to regularize the network so that it generalizes better.

---

### Author Response · Authors · 2020-11-22
**Responses to all reviewers**

We appreciate reviewers for their valuable and high-quality reviews, as well as for their constructive feedback. We addressed all individual reviewer’s comments.

*Summary*: We propose to model the last layer of a neural network by a combination of a hierarchical layer, and a spherical fully-connected layer, to regularize the network w.r.t. a given hierarchy. The Hierarchical layer encodes the hierarchy since it represents the adjacency matrix of the hierarchical tree, while the spherical layer forces the hyperplanes of similar classes to be close to each other. We formulate this reparametrization with a matrix-matrix product for efficient optimization with a standard pipeline. We also discuss the optimization of the neural network with the Riemannian gradient descent and show some empirical improvement.

We now list, then address, major concerns raised by most of the reviewers, about 1) the extension to non-mutually exclusive hierarchy, 2) the experimental part of the paper, and 3) the hyperparameters “R0” and “Radius decay.”
We sincerely try to address major issues in the paper raised by the reviewers. We hope this will affect positively their vision of our paper.

#### **1) Extension to hierarchical graphs**
A concern shared by many reviewers is the extension to a mutually non-exclusive hierarchical structure. In the paper, we considered trees for simplicity. However, we believe it is possible to extend the idea to graphs by considering a hierarchical layer that encodes the adjacency matrix of a graph instead. This means we have to rethink a bit some aspects of the methods, but we believe that such an extension is possible (with some additional modifications).


#### **2) Experiments and empirical evaluation**
We do agree with the reviewers that the weaker side of our paper is the section with numerical experiments, but we also believe they are representative of the efficiency of the approach - with the addition of a negligible number of learning parameters, we improve substantially the network accuracy, thanks to our parametrization.

We point out that is hard to compare fairly with prior work in the field: we only slightly modify the neural network with the hierarchical layer, while other techniques usually reparametrize the entire network. Moreover, our reparametrization is rather flexible and can be combined with other approaches that also encode hierarchical information in the network.
The reviewers presented many suggestions to go further in the analysis of the benefits of our technique: For instance,
- [R2] proposed to use a soft regularization with the ell-2 norm rather than spheres.
- [R3] proposed to include the accuracy for super-classes or to analyze the difference in the feature distribution between the plain network and the reparameterized networks (using, for instance, feature space visualization techniques)
- [R1] also proposes numerous ways to improve the clarity of our section describing experiments - which will be implemented in the paper.
***🠆 Please see responses below (https://openreview.net/forum?id=hbzCPZEIUU&noteId=wBTzL9kHwMa) to find those experimental results.***



#### **3) Hyperparameters**
We noticed that the reviewers have some questions about the hyper-parameter R0 (the initial radius of the sphere) and the “Radius decay.”
The initial radius R0 plays a little role in the performance, as changing R0 (says, doubling it) scales the entire last layer (i.e., double all the neural network outputs). We have seen that, experimentally, changing R0 does not play a role in the network accuracy and can be safely fixed to R0=1.
The radius decay parameter is indeed a hyper-parameter that needs to be tuned (see, for instance, table (4).  The reason we add radius decay is well summarized by R3: “labels in finer-grained level should be modeled with less capacity.” We noticed that the network performance is not too sensitive w.r.t. the radius decay parameter: for instance, for tiny-ImageNet in table 4, the network accuracy drops only by 0.5% if we set the radius decay at 0.8 instead of 1 (the best value we found). We agree with the reviewers that knowing the radius decay in advance is hard and left as an open question - however, this is the only parameter we need to tune in the model, and a good approximation of its optimal value can be easily found by cross-validation on a smaller network.

We hope this answer successfully the reviewer's concerns.

---

> ### Author Response · Authors · 2020-11-23
> **Newly added experiments**
>
> We added new experiments to address reviewer’s all comments on the experimental section regarding 1) (learnable) radius decay [R1, R2, R3, R4], 2) random hierarchy (usefulness of the hierarchical structure) [R3, R4], 3) classification performance of superclass [R3], 4) visualization of the feature space [R3, R4]. Experimental results are added in Appendix of our revised manuscript (texts in blue). A summary of results is as follow:
>
> #### **1)	Learning radius decay** (using ResNet-18, please see C.5 and Table 5 in the revised manuscript in detail)
>
> A learnable radius without constraints does not outperform the classification performance using the predefined radius decay (numbers shown in parentheses).
>
> |  Dataset   |  Hierarchy  | +Manifold |  +Riemann|
> | :---------|:-----------:|:-----------:|:-----------:|
> | CUB200  | 53.40 (58.28) | 58.35 (60.42) | 58.24 (60.98) |
> | Cars | 81.52 (84.96)  |82.54 (84.74) | 82.40 (84.16) |
>
> #### **2) Learning with random hierarchy** (using ResNet-18, please see C.6 and Table 6 in the revised manuscript in detail)
>
> Methods using a random hierarchy show considerably degraded performance compared to that using a manually annotated hierarchical information (numbers shown in parentheses).
>
> |  Dataset |  Multitask |  Hierarchy  | +Manifold |  +Riemann|
> | :---------|:-----------:|:-----------:|:-----------:|:-----------:|
> | CUB200  | 47.55 (53.99) | 50.28 (58.28) | 56.96 (60.42) | 56.43 (60.98) |
> | Cars | 79.98 (82.85) |81.07 (84.96)   |82.02 (84.74) | 81.84 (84.16) |
>
> #### **3) Superclass categorization** (using ResNet-18, please see C.7 and Table 7 in the revised manuscript)
>
> Our proposed methods (Hierarchy, +Manifold, and +Riemann) outperform  a multitask (multilabel) classification based method.
>
> |  Dataset |  Multitask |  Hierarchy  | +Manifold |  +Riemann|
> | :---------|:-----------:|:-----------:|:-----------:|:-----------:|
> | CUB200  | 53.68 | 58.87 | 61.17 | 62.22|
> | Cars | 86.88 | 87.91 |91.23|90.97 |
>
>
>
> #### **4) Visualization of feature space** (please see C.8 and Figure 4, 5, and 6 in the revised manuscript)
>
> We observed a distribution of learned two-dimensional embedding vectors, which are input vectors of the last classifier layer, as [R3] suggested. Moreover, we observed a distribution of multidimensional embedding vectors used in our other experiments, by reducing their dimensions using different popular techniques (t-SNE and PCA). Embedding vectors (two-dimension) of our proposed methods are well separated (clustered) compared to that the baselines.
>
> We will add more results using more methods (e.g. ResNet-50) and using other datasets in the revision.

---

### Decision · Program_Chairs · 2021-01-07
**Final Decision**

**Decision:**

Reject

**Comment:**

This paper introduces a method for hierarchical classification with deep networks. The idea is interesting, and as far as I know novel: namely, the authors add a regularizer to the last layer in order to enforce a hierarchical structure onto the classifiers. The idea of placing spheres (with a fixed radius) around each classifier and forcing the child-classifiers to lie on these spheres is quite clever.
The reviewers have pointed out some concerns with this paper. Some had to do with terminology (which the authors should fix but which is no big deal), but the main weakness are the experimental results and the ablation study. The reviewers were not convinced that the optimization in the Euclidean space wouldn't be sufficient. A more thorough ablation study could help here.

This is the kind of paper that I really want to see published eventually, but right now isn't quite ready yet. If you make one more iteration (in particular adding a stronger ablation study) it should be a strong submission to the next conference. Good luck!